# MicroRNA Changes in Gastric Carcinogenesis: Differential Dysregulation during *Helicobacter pylori* and EBV Infection

**DOI:** 10.3390/genes12040597

**Published:** 2021-04-19

**Authors:** Christian Prinz, Kemal Mese, David Weber

**Affiliations:** 1Medizinische Klinik 2, Helios Universitätsklinikum Wuppertal, 42283 Wuppertal, Germany; david.weber@helios-gesundheit.de; 2Lehrstuhl Innere Medizin 1, University of Witten/Herdecke gGmbH, 42283 Wuppertal, Germany; kemal.mese@helios-kliniken.de; 3Institute of Virology, University of Göttingen, 37075 Göttingen, Germany

**Keywords:** microRNA dysregulation, gastric cancer, *Helicobacter pylori*, Epstein–Barr virus (EBV), bamHI-A region rightward transcript (BART)

## Abstract

Despite medical advances, gastric-cancer (GC) mortality remains high in Europe. Bacterial infection with *Helicobacter pylori* (*H. pylori*) and viral infection with the Epstein–Barr virus (EBV) are associated with the development of both distal and proximal gastric cancer. Therefore, the detection of these infections and the prediction of further cancer development could be clinically significant. To this end, microRNAs (miRNAs) could serve as promising new tools. MiRNAs are highly conserved noncoding RNAs that play an important role in gene silencing, mainly acting via translational repression and the degradation of mRNA targets. Recent reports demonstrate the downregulation of numerous miRNAs in GC, especially miR-22, miR-145, miR-206, miR-375, and miR-490, and these changes seem to promote cancer-cell invasion and tumor spreading. The dysregulation of miR-106b, miR-146a, miR-155, and the Let-7b/c complex seems to be of particular importance during *H. pylori* infection or gastric carcinogenesis. In contrast, many reports describe changes in host miRNA expression and outline the effects of bamHI-A region rightward transcript (BART) miRNA in EBV-infected tissue. The differential regulation of these miRNA, acting alone or in close interaction when both infections coexist, may therefore enable us to detect cancer earlier. In this review, we focus on the two different etiologies of gastric cancer and outline the molecular pathways through which *H. pylori-* or EBV-induced changes might synergistically act via miR-155 dysregulation to potentiate cancer risk. The three markers, namely, *H. pylori* presence, EBV infection, and miR-155 expression, may be checked in routine biopsies to evaluate the risk of developing gastric cancer.

## 1. Introduction

Gastric-cancer (GC) outcomes remain poor despite therapeutic advances. In Europe in 2012, an estimated 139,600 GC cases were diagnosed, with an incidence rate of 13.7/100,000 person years (age-standardized rate), and approximately 107,000 deaths, with a mortality rate of 10.3 per 100,000 person years [1]. In 2019, it was reported that these rates only slightly decreased within European countries [2]. European GC patients have an overall 5-year survival rate of only 30–40%, whereas Japan achieved a 70% survival rate [3]. European countries with high incidence and mortality rates include Germany, Italy, Spain, and Portugal, even though endoscopic monitoring is easily available in these countries [4]. Numerous factors contribute to these poor outcomes, including poor detection during endoscopies, limited surgical experience, and aspects of chemotherapy. Most importantly, the majority of tumors are detected in the advanced stages, resulting in poor prognoses and outcomes since GC is generally not highly sensitive to chemotherapy, a treatment that is only successful in certain patient subgroups.

Recent trials investigated neoadjuvant chemotherapy in GC. Most prominently, the Medical Research Council Adjuvant Gastric Chemotherapy (MAGIC) trial demonstrated a clear benefit of perioperative and postoperative chemotherapy [4]. However, although this protocol is a current treatment standard, the overall survival rate of patients treated under this regimen in Europe remains only at around 40–45%. An evaluation of the clinical response to chemotherapy revealed that not all patients benefit from neoadjuvant, perioperative, and postoperative regimens. It is, therefore, of enormous clinical value to identify markers from biopsies that could predict the chemotherapy response in patients with or without subsequent surgery.

Infections with *Helicobacter pylori* [5] and with Epstein–Barr Virus (EBV) [6] were correlated with GC development; therefore, it seems rational to examine biomarkers indicating malignant developments during these infections. In particular, microRNA (miRNA) dysregulation may be a promising tool [7]. Current evidence highlights that coinfection with EBV can potentiate injury in gastric diseases [8,9]. In the current review, we focus on the two different etiologies of GC and outline the molecular pathways through which both agents might act synergistically to potentiate cancer risk.

## 2. Dysregulation of miRNAs in GC: Significance and Critical Roles as Diagnostic or Prognostic Biomarkers in Biopsies and in Human Plasma

miRNAs affected by EBV and/or *H. pylori* infections may provide new biomarkers for GC. They were recently associated with cancer development and progression in other gastrointestinal (GI) cancers. miRNAs recognize specific base pairings in mRNA and bind to them, thereby regulating gene translation and protein synthesis. They can also induce the deadenylation of mRNA, shortening the half-life of the corresponding mRNA molecule, and potentially triggering protein binding to certain DNA sections, modifying chromatin structure and gene expression [10]. Through these mechanisms, miRNAs play an important role in cell proliferation and differentiation, and in controlling apoptosis. On the basis of these molecular findings, the up- or downregulation of miRNAs may be associated with cancer development and progression, highlighting the potential use of miRNAs as predictive markers in various cancer stages [10,11,12,13].

Pioneering work by Lu et al. revealed the general downregulation of miRNAs in GI tumors when compared with that of normal tissue [12]. Moreover, their results demonstrated the identification of GI cancers using miRNA expression profiles, which were clearly superior to mRNA profiles, thus supporting the potential of miRNAs as diagnostic biomarkers. The investigators found that certain miRNAs were downregulated in cancer tissue, but upregulated in corresponding normal epithelia. However, they could not determine whether the miRNAs that were highly dysregulated in the gut-associated cluster (such as the Let-7 group, miR-192, miR-194, and miR-215) had specific diagnostic or prognostic value in GC. Moreover, the possible functional role of these miRNAs in such tumors remains unclear. Some of these miRNAs clearly have pleiotropic functional roles; for example, Let-7 microRNAs play a key role in development, stem cells, and cancer, and were detected in other cancer tissue types [13].

Recent studies described miR-21 expression [14,15] as both a diagnostic and a prognostic biomarker in GC, since the miRNA is highly upregulated in GC and significantly associated with tumor differentiation, local invasion, and lymph-node metastasis. The overexpression of miR-21 promotes GC cell growth, invasion, and migration in vitro, whereas miR-21 downregulation yields an inhibitory effect [16]. Zhang et al. reported that miR-21 plays a pivotal role in GC pathogenesis and progression through a process involving microsatellite instability [16]. The high miR-21 expression in GC is similar to the findings of Yan et al. in breast cancer [17]. Another study found that miR-370 downregulation is highly associated with more advanced nodal metastasis and a higher clinical stage of GC, acting via the inhibition of transforming growth factor-β receptor II expression [18].

Recent studies attempted to determine the value of certain miRNAs in human plasma as diagnostic markers for early-stage GC, including miR-17, miR-25, and miR-133b [19], miR-425-5p, miR-1180-3p, miR-122-5p, miR-24-3p, and miR-4632-5p [20]. A previous report that was inconsistent with previous data found miR-185, miR-20a, miR-210, miR-25, and miR-92b to be dysregulated in plasma from GC patients [21]. Overall, on the basis of the contradictory results, available data must be interpreted with caution. Discrepancies may be due to the complexity of human gene regulation in blood; however, all studies were hampered by a small sample size.

## 3. Importance of miRNAs in GC: Critical Roles in Tumorigenesis and Tumor Spreading, and as Molecular Targets

Numerous studies investigated dysregulated miRNA expression during GC development, invasion, and metastasis. In most tumors, potential oncogenes may become activated due to the downregulation of miRNAs that normally function in the repression of translated products, including miR-375. In GC cells, miR-375 is downregulated, allowing for the increased activation of PDK1 (a kinase with antiapoptotic effects) and the JAK2 oncogene, which is a process that leads to cancer-cell proliferation, migration, and invasion [22,23,24]. However, miR-375 cannot be regarded as a specific marker for GC, as it was recently found to be a biomarker that differentiates rectal cancer from colon cancer, potentially interfering with IGF-2 expression [25].

Studies showed that miR-181c overexpression causes the decreased growth of GC cell lines. Moreover, miR-181c may be downregulated in GC and play an important role through its target genes, such as NOTCH4 [26]. Recent studies indicate that NOTCH4 signaling is essential for vascular morphogenesis [27].

Recent reports also demonstrated the downregulation of miR-22, miR-145, and miR-490 in GC, and these changes seem to promote cancer-cell invasion and tumor spreading. In one study, miR-22 upregulation led to a decrease in GC cell migration and invasion through targeting oncogenic gene Sp1, while miR-22 downregulation allowed for increased infiltration [28]. Additionally, miR-145 reportedly suppresses cancer spreading through the inhibition of N-cadherin protein translation, acting via the downregulation of matrix metallopeptidase (MMP)-9 [29]. GC also exhibits downregulated miR-490-3p expression, which normally exerts growth- and metastasis-suppressive effects in cell lines by targeting SMARCD1 [30]. In turn, SMARCD1 expression is upregulated in the gastric tissue of patients with GC, and high SMARCD1 expression was associated with shorter patient survival independent of TNM staging [30]. GC also shows significantly decreased expression of miR-206 [31], which is a potential tumor suppressor acting via cyclin D2 (CCND2), a protein that plays a key role in controlling the cell cycle. MiR-206 suppresses GC cell proliferation, and reduces cell growth and colony-forming abilities [31]. Additionally, miRNA-331-3p is a potential tumor suppressor in GC that directly targets E2F1, a transcription factor with important functions in cell-cycle regulation, mediating the G1-to-S phase transition [32]. Furthermore, miR-107 may exert a tumor-suppressor function by directly targeting CDK6, a protein kinase that increases cell growth and blood-vessel supply. The ectopic expression of miR-107 inhibits proliferation, induces G1 cell-cycle arrest and blockades the invasion of GC cells [33].

In other tumor types, miRNAs are reportedly upregulated and exert different carcinogenic functions. For example, miR-221 and miR-222 are upregulated in GC cells, affecting cell growth and invasion possibly via the direct modulation of phosphatase and tensin homolog (PTEN) [34]. PTEN interacts with the PI3K–Akt signaling pathway by dephosphorylating PIP2, PIP3, and protein kinase B, thereby counteracting the antiapoptotic PI3K–Akt signaling pathway and inhibiting cell proliferation. Additionally, miR-616 acts through the PTEN/mTOR signaling cascade, thereby activating angiogenesis and tumor invasion through an increase in epithelial–mesenchymal transition [35]. Accordingly, miR-21 inhibition reportedly upregulates PTEN expression [35], indicating that PTEN may be a specific target gene for GC initiation and development.

A remarkably high number (15–30%) of gastric tumors exhibit microsatellite instability (MSI), which is characterized by the accumulation of mutations, namely, at repetitive sequences (microsatellites) due to a defective DNA mismatch repair system (MMR) [36] or mutations in different genes involved in DNA damage response (e.g., ATR or CHK1) [37]. Appropriate cellular responses to DNA damage and the integrity of DNA repair systems are critical for maintaining genomic stability, which is lost in many gastrointestinal tumors including GC. Table 1 presents an overview of miRNAs that are upregulated and downregulated in GC.

## 4. Roles of miRNA Dysregulation in *H. pylori*-Induced Gastric Adenocarcinoma

*H. pylori* is classified as a Type I class carcinogen due to the overwhelming scientific evidence that this bacterium colonizes the gastric mucosa in the presence of acid, leading to chronic gastric inflammation and gastric carcinogenesis. Several guidelines strongly recommend the eradication of this bacterium at the chronic gastric infection stage to prevent GC development [5]. However, true predictors and an exact time point of “no return” for a malignant transformation over decades remain to be determined.

miRNAs are a promising tool for addressing this important clinical question. *H. pylori* infection leads miR-210 to undergo CpG island methylation, thus inducing STMN1 expression [39]. The let-7 family members also exhibit involvement in *H. pylori*-induced cancer development. In vitro and in vivo data confirmed that certain miRNAs act as tumor suppressors in the GC development process, including Ras, Myc, and HMGA-2 [10,13,45]. The expression of Let-7c was assessed in biopsies from different stages of gastric inflammation and cancer, including *H. pylori*-related gastritis, atrophy, and intestinal metaplasia. Findings show let-7c dysregulation in the different stages, with increasing downregulation through the different stages of gastritis to atrophic and metaplastic gastritis, neoplasia, and invasive GC. Moreover, following *H. pylori* eradication, let-7c expression was significantly increased [46]. The downregulation of Let-7c was also observed in a mouse model following inoculation with *H. pylori*. Overall, available data suggest that early phases are characterized by let-7c dysregulations [46]. A similar function was detected for let-7b miRNA [47]. The expression of let-7b was reportedly decreased in gastric adenocarcinoma, with tumor stage, and in lymphatic metastasis. Collagen triple helix repeat containing 1 (Cthrc1) was described as a potential direct target of let-7b since Cthrc1 is significantly upregulated in GC. These findings suggest that let-7b may directly act as a tumor-suppressor gene in GC [47].

Investigations in *H. pylori-*infected patients show that low expression levels of miR-375 and miR-106b were correlated with inflammation scores and colonization density [48]. MiR-375 reportedly targets the Janus kinase JAK 2 [23]; thus, its downregulation may be responsible for increased tumor growth. In a recent study, *Helicobacter* lipopolysaccharide (LPS) led to the downregulation of miR-375 and miR-106b, which are associated with MDM2 expression, yielding JAK1 and STAT3 dysregulation, which serve as downstream target genes of the two miRNAs. These new targets within the crucial carcinogenic process seem to be especially associated with infection with the Type 1 cagA+ strains of *H. pylori*, following the tyrosine phosphorylation of JAK1, JAK2, and STAT3. Thus, the dysregulation of miR-375 and miR-106b during the *H. pylori* LPS-induced signal cascade may substantially alter JAK1/JAK2 and STAT3 signaling [49]. This suggests that STAT3 levels in the gastric mucosa may be of further prognostic impact [50].

The expression of miR-146a is also of particular interest in the pathway of *H. pylori*-induced gastric carcinogenesis, and it is reportedly dysregulated in GC [42,51]. This miRNA inhibits the cytokine-induced inflammatory responses during infection [51], and was also described and investigated in *H. pylori*-infected gastric tumors. In one study, the overexpression of miR-146a was associated with the progression of gastric tumors, with greater stages and lymph-node metastasis; furthermore, miR-146a expression was found to be independent of *H. pylori* infection [42]. In other studies, the treatment of immune cells with bacterial LPS resulted in the induction of both miR-146a and miR-155 [52], outlining a direct effect of *H. pylori* on the dysregulation of these miRNAs.

### Importance of EBV Infection for GC Development

Certain types of GC and nasopharyngeal carcinoma are dependent on EBV infection [53,54,55]. EBV-infected GC constitutes a latent Type I infection and does not express BHRF miRNAs [56,57]. In contrast, EBV-associated nasopharyngeal cancer is a latent Type II infection [57]. Moreover, while EBV infection is commonly present in patients with nasopharyngeal carcinoma, it is found in only 8–10% of GCs [6]. EBV-positive tumors occur at both sites of GC, i.e., in both proximal and distal types [56]. Previous studies determined differential localization in patients with EBV-positive vs. EBV-negative GC [58]; however, sample sizes were not representative. Regarding EBV and miRNA, many reports described changes in host miRNA expression and particularly the effects of bamHI-A region rightward transcript (BART) miRNA [59]. In EBV-induced GC, targets of BART miRNAs were described. For example, mutations were detected in PIK3CA (80%), which regulates the PI3K/Akt signaling pathway, and in ARID1A (55%), which has DNA helicase and ATPase functions [52,53,54]. On the other hand, chromosomal abnormalities are not common in EBV-associated cancer [60,61,62].

EBV presence differs in the acute- and latent-infection periods. During the latent-infection period, few EBV proteins are expressed, but there is high expression of EBV miRNAs, such as BART miRNAs [60,61,62]. EBV-coded miRNA-BARTs are expressed in B-cell lymphoma (latency I), NK/T cell lymphoma (Latency I), Hodgkin’s disease (Latency II), and in EBV-associated PTLD (Latency III). In EBV-infected cells, three miRNAs are produced in the BHRF1 locus; these miRNAs flank the BHRF1 open-reading frame encoding a Bcl2 viral homolog, and are especially expressed in EBV Latent III infections [60,63,64]. Furthermore, 22 miRNAs [57] can be divided into two subsets of BART-miRNAs [57]. BART Group 1 members include miRNA-BART1, 3–6, and 15–17, while BART Group 2 members include miR-BART-18–21 and miR-BART-7–14 [61]. BART Group 2 members (including miRNA-BART18–3p, 7–5p, 10–3p, 10–5p, 11–3p, 13–5p, and 14–5p) are also expressed in Type III latent EBV infections [65,66]. Reportedly, miRNA-BART2-5p and miRNA-BART2-3p are the rear members of these two subgroups [67]. 

In EBV-associated GC, tumor cells show latent infections with EBV I and express the small RNA encoded in EBV EBNA1 [68]. In EBV-transformed B cells and infected GC cells, miRNA-BART16 directly targets and regulates the CREB-binding protein, which is an important transcriptional coactivator in the interferon signaling pathway (IFN). Therefore, miRNA–BART16 facilitates the establishment of latent EBV infection and promotes viral replication by diminishing the antiviral action of interferon-α [69]. Furthermore, EBV-associated miRNAs have important impact on the production of interleukin-1 (IL-1) [70].

## 5. Research Perspective Given Functional Importance of *H. pylori*- and EBV-Induced Changes in GC Acting Synergistically

Various studies investigated the correlation between EBV and *H. pylori* infection of the stomach, and infection with both pathogens is associated with stomach-cancer development. The spectrum of miRNAs induced in both infections seems to differ widely between the two types of infection in vivo and, especially, in vitro. However, it seems rational from a clinical perspective that both infections might act together to promote further cancer development.

Given the importance of EBV in GC and the possible interactions with *H. pylori* infection, presently available data suggest the importance of using molecular methods to directly evaluate the presence of both *H. pylori* and EBV in biopsy samples [71]. Moreover, current data suggest that the two different etiologies may synergistically act to induce GC, and that miR-155 may play a special role in this process.

With regards to the carcinogenic potential during *H. pylori* infection, the induction of miR-155 by *H. pylori* is dependent on the Type IV secretion system and is related to the presence of the cagA pathogenicity island [72]. These findings may explain why only the so-called Type 1 strains are associated with GC development. Recent data also indicate that miR-155-5p overexpression plays a notable role in *H. pylori*-induced inflammation and carcinogenesis [73]. Furthermore, miR-155 overexpression was observed in GC biopsies upon *H. pylori* infection and in gastric adenocarcinoma [14,15,52,74]. Interestingly, miR-155 regulation is dependent on the activator protein 1 (AP-1) pathway in B cells and seems to enable the persistence of EBV in immune cells [14,15]. Thus, this molecule may be of particular importance for evaluating GC risk.

Infection with *H. pylori* and EBV may lead to the infiltration of regulatory T cells; in turn, increased Foxp3 expression reportedly controls miR-155 expression in T cells [14,15,52,74]. Additionally, bacterial LPS exposure induces miR-155 expression in immune cells [52,75]. Thus, this interesting miRNA could be a critical determinant of GC development during the initial steps of tumor formation and may serve as a diagnostic biomarker in early cancer stages.

On the basis of available data (shown in Table 2), coinfection with *H. pylori* could activate EBV-infected B cells, and miR-155 plays a pivotal role in this process [52]. B cells express the CD21 receptor, a known EBV receptor. In turn, infected B cells might infiltrate the chronically inflamed tissue of the gastric epithelial cells, and this cell-to-cell binding between B cells and gastric epithelial cells may result in viral entry into gastric epithelial cells, permitting the initiation of its oncogenic potential. Further details of the mechanism underlying this phenomenon remain unclear, but miRNA activation might be involved in additional relevant pathways during these processes [76,77]. The targeted screening of EBV-associated and *H. pylori*-induced miRNAs offers the possibility of establishing more individualized therapy. For this purpose, the entire spectrum of miRNAs (and particularly, the total amount of miR-155 expression) could be examined at the cellular level in *H. pylori*- and EBV-associated tumor diseases, and the tumorigenic potential may be deducted from those observations. Table 2 presents an illustration of this process.

## Figures and Tables

**Table 1 genes-12-00597-t001:** Overview of microRNAs (miRNAs) upregulated and downregulated in gastric cancer (GC) and potential targets.

Downregulated miRNAs in GC and Potential Targets		Importance
miR-22	Downregulation seems to activate oncogenic gene Sp1 [28].	Promotes cell migration and invasion in GC cell lines.
miR-30b	Downregulation targets plasminogen activator inhibitor [38].	Reduced apoptosis of cancer cells.
miR-107	Tumor-suppressor functions by targeting CDK6 [33].	Expression inhibits proliferation and induction of G1 cell cycle arrest; promotes invasion of GC cells.
miR-145	Inhibits N-cadherin protein translation and acts via downregulation of matrix metallopeptidase 9 [29], downregulation of potential tumor suppressor [30].	Hypermethylation leads to miRNA downregulation, which is associated with increased tumor growth and spreading.
miR-181c	Targets genes such as NOTCH4 and KRAS [26].	Downregulation in tumors leads to increased growth of GC cell lines.
miR-206	A potential tumor suppressor targeting cyclin D2 (CCND2) [31].	Countereffect: miR-206 suppresses GC cell proliferation, reducing cell growth and colony-forming abilities.
miR-210	Alters CpG methylation [39].	Downregulation enhances growth of GC cells.
miR-331-3p	A potential tumor suppressor in GC; directly targets E2F1 [32].	Downregulation enhances growth of GC cells.
miR-370	Inhibits expression of transforming growth factor-β receptor II [18].	Downregulation associated with more advanced nodal metastasis and a higher clinical stage of GC.
miR-375	Acts as tumor suppressor by targeting JAK2 oncogene [22,23,24].	Downregulated in GC cells, reduces cell viability via caspase-mediated apoptosis pathway through downregulation of PDK1.
miR-422	Targets pyruvate dehydrogenase kinase 2 (PDK 2) [40].	Downregulation of miR-422 enhances tumor growth.
miR-490	Downregulation of potential tumor suppressor [30].	Enhances tumor growth.
miR-490-3p [30]	miR-490-3p suppresses growth and metastasis in cell lines by targeting SMARCD1 [30].	Prognostic biomarker due to close correlation with shorter patient survival independent of TNM staging.
miR-1228	Targets macrophage migration inhibitory factor [41]^.^	Downregulated in GC; impairs the proangiogenic activity of GC cells.
**Upregulated miRNAs in GC** **and Potential Targets**		**Importance**
miR-21	Affects microsatellite instability (MSI); significantly associated with poor tumor differentiation [16].	miR-21 overexpression promotes GC cell growth, invasion, and migration in vitro; prognostic marker for local invasion, and lymph-node metastasis.
miR-107	Upregulated in gastric cancer [12].	Continuing upregulation from normal mucosa, adenoma, and cancer.
miR-146a	Gastric tumors and chronic gastric inflammation show miR-146a overexpression [42,43].	miR-146a dysregulation may promote gastric tumorigenesis and metastasis.
miR-221 and miR-616-3p	Direct modulation of PTEN [34].	Upregulated in GC cells, affecting angiogenesis and invasion.
miR-222	Inhibition of hepatocyte growth factor and activator-inhibitor type 1 protein expression [44].	Overexpression of miR-221 and miR-222 promotes cell proliferation and migration.
miR-300	Upregulated in gastrointestinal cancer [12].	Continuing upregulation from normal mucosa, adenoma, and cancer.

**Table 2 genes-12-00597-t002:** Overview of diagnostic and prognostic relevance of certain miRNAs during *Helicobacter pylori* and Epstein–Barr virus (EBV)-induced gastric inflammation, and gastric carcinogenesis, outlining that most miRNAs are not involved in both EBV and *H. pylori* infection, and only one miRNA (miR-155) seems to be affected by both etiologies. Three miRNAs of the let-7 group, and miR-106b and miR-145, are dysregulated both during the *H. pylori*-induced process of inflammatory changes and during gastric-cancer progression, and may thus be considered to be new and interesting biomarkers during the continuing and chronic disease.

Dysregulated miRNA during *H. pylori* Infection	Roles and Functions of Involved miRNAs	Dysregulated miRNA during EBV Infection	Roles and Functions of Involved miRNAs
		BamHI-A region rightward transcript (BART)-miRNA dysregulation reported during Type 1 latent infections relevant to GC development.	BART-miRNAs are divided into two subsets [61]; miRNA-BART2-5p and miRNA-BART2-3p are reported to be the rear members of these two subgroups, and may affect bcl2-dependent pathways [67].
		miRNA-BART16 [69,70]	Targets and regulates CREB binding, a transcriptional coactivator in the interferon signaling pathway (IFN); facilitates latent EBV infection by inhibiting IFN Type I-induced antiviral response.
miR-155 [74]	Involved in immune response; bacterial lipopolysaccharide (LPS) exposure induces miR-155 expression in immune cells; potential role as diagnostic marker; increased expression upon *H. pylori* infection and in gastric adenocarcinoma.	miR-155 [14,15]	Dysregulation of miR-155 enables persistence of EBV in immune cells; miR-155 regulation is dependent on activator protein 1 (AP-1) pathway in B cells [14,15].
Let-7b [47]Let-7c [46]	Correlates with inflammatory process, tumor stage, and lymphatic metastasis in *H. pylori*-induced gastric adenocarcinoma.		
miR-106b [49]	Inhibition of miR-106b associated with STAT3 signaling, a key molecule during *H. pylori*-induced immune responses; varies with inflammatory scores and cancer development.		
miR-146a [42,43]	Gastric tumors and chronic gastric inflammation show miR-146a overexpression.

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
