# Peer review of "MicroRNA Changes in Gastric Carcinogenesis: Differential Dysregulation during Helicobacter pylori and EBV Infection"

_genes, 2021, doi:10.3390/genes12040597_

Round 1

Reviewer 1 Report

This is a valuable paper that summarizes the microRNA changes  associated with  gastric carcinogenesis in H. pylori and EBV infection. It makes sense to consider biomarkers that suggest the development of gastric cancer due to these infections, since dual infections have implications, and to focus on microRNA as a tool for early detection of both infections. Although the spectrum of microRNAs in each of infections varies widely and a myriad of microRNA groups is intricately related to the carcinogenesis process of gastric cancer, this paper does an excellent job of summarizing the list of microRNAs that exhibit upregulation and down regulation in gastric cancer in an easy-to-understand manner.  The most significant finding of this paper is that there are two molecular pathways associated with gastric carcinogenesis, and that the two pathways work in concert through the expression of miR-155. The contribution of miR-155 expression in regulatory T cells to the mechanism, by which the two viruses work together to induce gastric cancer is of academic interest.

Author Response

Hello, and thank you very much for the reply. Due to the reviewers response, we have modified the abstract slighty, and added  new references in several sentences of the discussion section where a citation was missing. We hope that with the changes made, the manuscriot is now suitable for publication. Greetings, Christian Prinz

Reviewer 2 Report

  1. The abstract does not match what has been described in the main text. From abstract, the author state "In the current review, we focus on the two different aetiologies of gastric cancer, and outline molecular pathways through which H. pylori- or EBV-induced changes might synergistically act via miR-155 dysregulation to potentiate the risk of cancer." But H.pylori, EBV, and miR-155 were only talked about in the last part.
  2. The authors make some claims without solid reference support. Like in line 279 "microRNA activation might be involved in additional relevant pathways during these processes."
  3. Line 202-204, "In one study, overexpression of miR-146a was reduced with progression of gastric tumours, with greater stages and lymph node metastasis; however, miR-146a expression was independent of H. pylori infection" is confusing.
  4. Line 213-215, the reference is missing. (There are many place like this with statements without reference.)

Author Response

Hello, and thank you very much for the reply. Due to the reviewers response, we have modified the abstract slighty, and added  new references in several sentences of the discussion section where a citation was missing. We hope that with the changes made, the manuscriot is now suitable for publication.

Changes were made in the abstract and in the discussion section as follows:

Abstract – Question 1: The abstract does not match what has been described in the main text. 

Response to Question 1: The Title and the corresponding text in the manuscript has been changed as follows:

…NEW TITLE:  "MicroRNA changes in gastric carcinogenesis: differential dysregulation during Helicobacter pylori and EBV infection"

… (abstract line 23-24)... Recent reports demonstrate the downregulation of numerous microRNAs, especially miR-22, miR-145, miR-206, miR-375 and miR-490 in GC, and these changes seem to promote cancer cell invasion and tumour spreading.  In particular, dysregulation of miR-106b, miR-146a, miR-155, and the Let-7b/c complex seems to be of particular importance during H. pylori infection as well as gastric carcinogenesis.

Discussion - Question 2

The authors make some claims without solid reference support. Like in line 279 "microRNA activation might be involved in additional relevant pathways during these processes."

Answer to Q2:  

The following sentence and two new citations have been added. Changes are marked with underlines in the text as follows:

„In turn, the infected B cells might infiltrate the chronically inflamed tissue of the gastric epithelial cells, and this cell-to-cell binding between B cells and gastric epithelial cells may result in viral entry into gastric epithelial cells, permitting the initiation of its oncogenic potential. Further details of the mechanism underlying this phenomenon remain unclear, but microRNA activation might be involved in additional relevant pathways during these processes (ref. 76,77).“

R76:   Zhang Z, Li Z, Li Y, Zang A. MicroRNA and signaling pathways in gastric cancer. Cancer gene therapy. 2014;21(8):305-316.

  1. Krump NA, You J. Molecular mechanisms of viral oncogenesis in humans. Nature reviews Microbiology. 2018;16(11):684-698.

Discussion, Question 3: Line 202-204, "In one study, overexpression of miR-146a was reduced with progression of gastric tumours, with greater stages and lymph node metastasis; however, miR-146a expression was independent of H. pylori infection" is confusing.

Answer to Q3: the sentence has been altered as follows:

„In one study, overexpression of miR-146a was associated with progression of gastric tumours, with greater stages and lymph node metastasis; and in this study, miR-146a expression was independent of H. pylori infection42. In other studies, the treatment of immune cells with bacterial lipopolysaccharide resulted in the induction of both miR-146a and miR-15552, outlining a direct effect of H. pylori on the dysregulation of these microRNAs. „

Discussion, Question 4: Line 213-215, the reference is missing. (There are many place like this with statements without reference.)

Answer to Q4: the following sentence has been added, a total of seven references has been added tot he manuscript as follows:

Certain types of GC and nasopharyngeal carcinoma are dependent on EBV infection 53-55. EBV-infected GC constitutes a latent type I infection and does not express BHRF miRNAs56,57. In contrast, EBV-associated nasopharyngeal cancer is a latent type II infection57. Moreover, while EBV infection is commonly present in patients with nasopharyngeal carcinoma, it is found in only 8–10% of gastric cancers6. EBV-positive tumours occur at both sites of GC, i.e. in both proximal and distal types 56. Previous studies have determined differential localization in patients with EBV-positive vs. EBV-negative GC 58;

And also:

Infection with H. pylori as well as EBV may lead to the infiltration of regulatory T cells and, in turn, increased Foxp3 expression reportedly controls miR-155 expression in T cells14,15,52,74. Additionally, bacterial LPS exposure induces miR-155 expression in immune cells52,75. Thus, this interesting microRNA could be a critical determinant of GC development during the initial steps of tumour formation, and may serve as a diagnostic biomarker in early cancer stages.

As desired by the reviewer, every microRNA mentioned in the text and especially in table 1 is now equipped with a corresponding reference citation.

We hope that with the changes made, the manuscript is now ready for publication.

Greetings, Christian Prinz, Germany
